# The burden and care cascade in young and middle-aged patients with diabetes hypertension comorbidity with abdominal obesity in India: A nationally representative cross-sectional survey

**Saurav Basu**[1], **Vansh Maheshwari**[1], **Mansi Malik**[1], **Kara Barzangi**[2]*, **Refaat Hassan**[2]

**1** Indian Institute of Public Health - Delhi, Public Health Foundation of India, Haryana, India, **2** University of Cambridge, Trinity Ln, Cambridge, United Kingdom

* kaom2@cam.ac.uk

**Data Availability Statement:** The NFHS-5 survey data is available free of charge on request from the DHS program portal (https://dhsprogram.com/

## Abstract

We ascertained the burden, determinants, and care cascade in the young and middle-aged patients having co-existing hypertension (HTN), Diabetes Mellitus (DM), and abdominal obesity in India from a secondary data analysis of nationally representative data. The study examined cross-sectional data from the National Family Health Survey (NFHS-5) conducted in India from 2019 to 2021 in 788974 individuals aged 15–49 years including 695707 women and 93267 men. The weighted prevalence of DM-HTN comorbidity with high waist circumference in the sample was 0.75% (95% CI: 0.71 to 0.79) including 46.33% (95% CI: 44.06 to 48.62) newly diagnosed cases detected for HTN and high blood sugars. The weighted prevalence of Metabolic syndrome as per NCEP ATPIII criteria was found to be 1.13% (95% CI: 1.08 to 1.17). Only 46.16% existing cases were treated with both anti-diabetes and antihypertensive medication (full treatment), while 34.71% cases were untreated. On adjusted analysis, increasing age, females, higher wealth index, high fat diet, obesity and comorbidities were significantly associated with having DM-HTN comorbidity along with high-waist circumference. More than half of young and middle aged-population in India with DM-HTN-abdominal obesity triad are not initiated on treatment for DM and HTN comorbidities, while a majority of the previously diagnosed cases have uncontrolled blood pressure and poor glycemic control. The poor cascade of care for DM and HTN in these high-risk group of patients may substantially increase their risk for early progression and severity of microvascular and macrovascular complications especially cardiovascular disease.

## 1. Introduction

The Global Action Plan for the Prevention and Control of Noncommunicable Diseases (NCDs), aims to halt by 2025 the increasing burden of diabetes mellitus (DM) and reduce the relative prevalence of hypertension (HTN) by 25% [1]. NCDs account for 71% of the total

data/dataset/India_Standard-DHS_2020.cfm?flag=0). File names for men and women datasets are IAMR7EDT.ZIP and IAIR7EDT.ZIP respectively.

**Funding:** The authors received no specific funding for this work.

**Competing interests:** The authors have declared that no competing interests exist.

global mortality and 15 million premature deaths before the age of 70 years, with the WHO setting the goal of reducing premature mortality from NCDs by one-third by 2030 [1, 2]. The number of deaths from NCDs in India in 2017 was estimated to be around 4.7 million, accounting for 49% of all-cause mortality with cardiovascular diseases (23%) being the leading cause [3]. The leading risk factors driving the burden of cardiovascular diseases worldwide include diabetes mellitus (DM), hypertension (HTN), and obesity [4, 5]. Diabetes is a group of disease characterized by elevated blood glucose levels due to absolute or relative insulin deficiency, with 74 million individuals in India living with DM as of 2021 [6]. Hypertension, signifying increased force of blood against arterial walls has an estimated prevalence of 24% of men and 21% of women as per a nationally representative cross-sectional survey in the 15–49 aged population of India [7, 8]. Abdominal obesity is a condition of excessive fat deposition around the abdominal region signifying insulin resistance and ectopic fat accumulation with elevated adipokine levels that substantially increases the risk of multiple adverse health conditions including T2DM, hypertension, CVD, nephrotic syndrome, and even cancer [9].

Metabolic syndrome is a cluster of conditions including high blood pressure, high blood glucose, increased abdominal fat, and raised cholesterol level that accentuates by more than twofold the risk of acute cardiovascular events. The National Cholesterol Education Program (NCEP) Adult Treatment Panel III (ATP III) characterizes the criteria of metabolic syndrome as the presence of three or more of the following five conditions: waist circumference exceeding 40 inches for men or 35 inches for women, blood pressure exceeding 130/85 mmHg, fasting triglyceride levels (TG) over 150 mg/dl, fasting High-density lipoprotein (HDL) cholesterol level below 40 mg/dl for men or 50 mg/dl for women, and fasting blood sugar over 100 mg/dl [10].

A recently published systematic review estimated 30% prevalence of Metabolic syndrome among the adult population in India with six times higher odds of the condition in obese individuals [11, 12]. Regardless of prior histories of cardiovascular events, people with Metabolic syndrome have an average four-fold increased risk of getting a stroke or myocardial infarction and a two-fold increased risk of dying from a comparable event [13].

Addressing the challenge of Metabolic syndrome in India is essential towards reducing the risk of NCD related complications with high morbidity, mortality, and associated socioeconomic health costs. Lifestyle modifications through healthy diet and exercise can reverse prediabetes, early-stage hypertension, and abdominal obesity thereby reducing the risk of its complications [14, 15]. Furthermore, in existing patients with DM and/or HTN with abdominal obesity, adherence to medications apart from healthy lifestyle and self-care is crucial for maintaining controlled blood glucose and reduced blood pressure levels that reduce the risk of complications [16, 17]. However, evidence from national level surveys from India suggest that most patients with DM and/or HTN fail to achieve optimal blood pressure and glycemic control that is associated with reduced quality of life and poor health outcomes [18, 19]. Patients with multimorbidity conditions as in Metabolic syndrome are subjected to complex pharmacological regimes that is associated with reduced medication adherence due to polypharmacy and regimen complexity and potential drug interactions and associated side effects [20, 21].

Care cascade for a chronic disease indicates the sequential steps a person would take from the stages of screening, diagnosis, treatment initiation, and achievement of target health outcomes. Analysis of these cascade can help to generate quantitative evidence on the gaps in the existing framework for delivery of care and the areas of concern where quality of care warrant improvement [22]. However, till date, evidence from nationally representative survey data has not been utilized to assess the quality and effectiveness of the care cascade especially in young and middle-aged patients with Metabolic syndrome in India.

We therefore conducted this study with the objective of ascertaining the burden, determinants, and care cascade in the young and middle-aged patients having co-existing HTN, DM, and abdominal obesity in India.

## 2. Methods

### 2.1 Study design and data source

The present study utilized data collected from the nationally representative cross-sectional survey National Family Health Survey (NFHS-5) conducted in India from 2019 to 2021. The survey was carried out with the support of the Ministry of Health and Family Welfare (MoHFW), coordinated by the International Institute for Population Sciences (IIPS) in Mumbai, and executed through collaboration among various survey organizations and Population Research Centres. A two-stage stratified sample is utilized in the NFHS-5. The probability proportional to size (PPS) sampling method was used to select the primary sample units (PSUs), which were villages in rural areas and Census Enumeration Blocks (CEBs) in urban areas. Detailed information pertaining to the sampling, data collection and survey instruments can be found in the NFHS India Report [8].

### 2.2 Study participants

The study adhered to specific inclusion and exclusion criteria as outlined by the National Family Health Survey 5 (NFHS-5). Inclusion criteria comprised individuals residing within selected households or clusters in the surveyed areas, with age considerations depending on the survey's focus. Participation was contingent on obtaining informed consent and participants were required to be Indian citizens. Conversely, exclusion criteria included pregnant women (self-reported as yes or no), non-consent, non-residence within the selected households or clusters, non-cooperation, and unavailability during the survey period. This study included 788974 individuals (695707 women and 93267 men) aged 15–49 years who provided consent for blood pressure, anthropometric and random blood sugar (RBS) measurements during the survey.

### 2.3 Data collection and preparation of samples

The data collection for NFHS-5 utilized a comprehensive questionnaire encompassing thematic categories including socio-demographic information, lifestyle characteristics, and access to healthcare services. Questions were carefully crafted to ensure clarity and relevance across diverse populations. The questionnaire, crafted in 18 regional languages, was rigorously tested and pretested, then deployed via computer-assisted personal interviewing. To guarantee data quality, field staff underwent multiple training sessions before the final survey implementation. In addition to self-reported data, biomarker measurements were collected to enhance the accuracy of certain health indicators. These measurements included blood glucose, blood pressure and anthropometric measurements. Standardized protocols were followed to guarantee consistency in data collection across survey teams.

For blood glucose testing, the Accu-Chek Performa glucometer with glucose test strips was used to collect finger-stick blood samples. A previously undiagnosed individual was considered to have diabetes if the RBS ≥200mg/dl or ≥126 mg/dl if fasting for ≥8 hours during the survey. Blood pressure was measured for all women and men aged 15 and above using an Omron Blood Pressure Monitor to determine the prevalence of hypertension. Blood pressure measurements for each respondent were taken three times with an interval of five minutes between readings [8]. The average of the second and third measurements was taken and

participants with SBP of ≥140 mmHg and/or a DBP of ≥90 mmHg were regarded as hypertensive, while those already on antihypertensive medicines to regulate their BP were also considered hypertensive [23]. The waist circumference (WC) is a validated measurement of abdominal obesity [24]. The waist circumference measurement for assessment of abdominal obesity were done by using Gulick tapes [8].

## 2.4 Operational definitions

1. Previously diagnosed patients with DM: Those who said yes to "told they had high glucose levels on two or more occasions by a doctor or other health professional", or taking medications to lower blood glucose levels were considered as previously diagnosed.

2. Newly diagnosed patients with DM: Individuals who were not fasting and had RBS levels ≥200 mg/dl or ≥126 mg/dl if fasting for ≥8 hours during the survey, and responded 'no' to the following two questions:

   a. Told they had high glucose on two or more occasions by the doctor.

   b. Currently taking prescribed medicine to lower glucose levels.

3. Previously diagnosed patients with Hypertension: Those who said yes to "told they had high bp on two or more occasions by a doctor or other health professional", or taking medications to lower blood pressure were considered as previously diagnosed.

4. Newly diagnosed patients with Hypertension: Individuals who had an average of the last two blood pressure readings (BP) ≥140/90 mmHg on screening during the survey and responded 'no' to the following two questions:

   a. Told they had high BP on two or more occasions by the doctor.

   b. Currently taking prescribed medicine to lower BP.

5. Abdominal Obesity: Present if waist circumference ≥35 inches for women and ≥40 inches for men.

6. On Diabetes Treatment: Individuals who responded 'yes' to the following question: "Currently taking a prescribed medicine to lower blood glucose", were considered to be on treatment for hypertension.

7. On Hypertension Treatment: Individuals who responded 'yes' to the following question: "Currently taking a prescribed medicine to lower blood pressure", were considered to be on treatment for diabetes.

8. Metabolic syndrome was defined as per National Cholesterol Education Program (NCEP) Adult Treatment Panel III (ATP III) [10] as the presence of three or more of the following five criteria: waist circumference over 40 inches (men), or 35 inches (women), blood pressure over 130/85 mmHg, fasting triglyceride levels (TG) over 150 mg/dl, fasting High-density lipoprotein (HDL) cholesterol level <40 mg/dl (men), or 50 mg/dl (women), and fasting blood sugar over 100 mg/dl. Since, data on TG and HDL was not available in the NFHS, we defined metabolic syndrome as the presence of the remaining 3 conditions (S1 Table).

## 2.5 Outcome variables

1. Diabetes-hypertension comorbidity along with high waist circumference was the primary outcome variable of this study. Cases of DM and HTN included both previously diagnosed cases and newly diagnosed cases on screening during the survey. Subsequently, DM-HTN-high waist circumference cases were classified as either old cases when the participant was aware of their pre-existing DM and HTN status; and new cases when the survey participants were detected having DM or HTN on blood sugar and blood pressure screening.

2. Suboptimal glycemic was defined as RBS $\geq$180 mg/dl and for fasting $\geq$130 mg/dl [25]) and suboptimal blood pressure control was defined as BP $\geq$140/90 mmHg) as per the JNC-7 criteria [23].

3. Old (previous diagnosed) patients with DM-HTN comorbidity and high waist circumference were considered on full treatment if they were currently taking both anti-diabetes and anti-hypertensive medications, partial treatment were those on either anti-diabetes or anti-hypertensive medication, while no treatment were those patients on neither anti-diabetes nor anti-hypertensive medications.

## 2.6 Independent variables

Covariates included personal information such as self-reported age, sex, employment status, educational attainment, smoking status, level of physical activity, and marital status (individual-level variables) and wealth status, place of residence, and region (household level variables). Anthropometric measurements included waist circumference, weight (in kgs) and height (in cms). Body mass index (BMI) was calculated using the weight and height of participants and defined as: Underweight (<18.5 kg/m2), Normal (18.5–24.9 kg/m2), Overweight (25.0–29.9 kg/m2) and Obese ($\geq$30.0 kg/m2) [26]. Lifestyle information included physical activity, diet, additional comorbidities and media exposure. Information on physical activity was not directly available in the NFHS-5 dataset. Hence, occupation was considered as a surrogate to assess the physical activity levels. Any respondent whose work responsibilities involved physical activities were regarded as 'involved in an occupation with high physical activity'; otherwise, they were considered to 'involve less physical activity' i.e., Not working, clerical and sales jobs = involve less physical activity and other job types = involve high physical activity [27]. Diet was categorized into a high and low-fat diet based on previous literature [28, 29]. The high-fat diet consisted of individuals who said 'Yes' to either "Daily eggs", "Daily fish", "Daily chicken or meat" or "Daily fried foods" consumption. Whereas, the remaining participants were put in the Low-fat diet category. Additional Comorbidities was treated as a dichotomous variable where respondents were assigned to the 'No comorbidity' group if they responded 'no' to all the following questions: "do you currently have any heart disease?; do you currently have chronic respiratory diseases including asthma?; do you currently have a goitre or any other thyroid disorder?; do you currently have chronic kidney disease?" If the participant responded 'yes' to any one or more of the questions above, then they were grouped in the "one or more comorbidity" group. For media exposure, if the respondent said 'no' to the question: "Owns a mobile telephone" and said "not at all" to the question: "frequency of watching television/listening to the radio/ reading the newspaper" then the participant was considered to have "no media exposure". If the respondent said 'yes' to any of the questions mentioned above, then the participant was considered to "have media exposure".

## 2.7 Statistical analysis

Descriptive analysis was performed, wherein categorical variables were reported as frequencies and proportions, while continuous variables were reported as means and standard deviations (SD). Bivariate analysis was performed and the chi-square test was used to identify the relationship between dependent and predictor variables. Furthermore, we applied multinomial regression model to estimate the regression coefficients (relative risk ratio (RRR)) for all three treatment behaviours. Variables found to be significantly associated in the crude model were included in the adjusted multinomial model. The model assumptions regarding linearity in the logit function, multicollinearity, and the presence of outliers were assessed.

We utilized the modified Poisson regression approach [30] for assessing the determinants of previously diagnosed, incident and prevalent DM-HTN comorbid cases with high-waist circumference. Both crude and adjusted rate ratio (RR) were reported. The model was fit using a Generalized Linear Model (GLM) with the Poisson family and a robust error variance. Model fit was checked by Akaike Information Criterion (AIC) and the Bayesian Information Criterion (BIC).

A variable was considered to be statistically significant if the P-value was less than 0.05. Each estimate was accompanied by confidence intervals at the 95% level. The "svy" suffix was applied to perform weighted analysis to regulate the clustering effect. Data analysis was conducted using Stata version 15.1 (StataCorp, College Station, TX, USA).

## 2.8 Ethics approval

The ethical approval for NFHS 5 survey was obtained from ethics review board of the International Institute of Population Sciences, Mumbai, India. Written and informed consent was obtained from each respondent before participating in the survey, additionally, written informed consent was obtained from the parent/guardian of each respondent under 18 years of age. No separate ethical approval was required for this secondary data analysis, since the NFHS-5 dataset is an anonymous publicly available dataset with no identifiable information about the study participants. The data were accessed on the 15th of February 2023 for research purposes, and authors never had access to information that could identify individual participants during or after data collection.

## 3. Results

The current study included 788974 individuals aged 15–49 years (695707 women and 93267 men) in our analysis. The mean (SD) age of the participants was 30.64 (9.99) years (S2 Table). The weighted prevalence of DM-HTN comorbidity with high waist circumference was 0.75% (n = 4644, 95% CI: 0.71 to 0.79) in the sample including 46.33% (n = 2070, 95% CI: 44.06 to 48.62) newly diagnosed cases detected on screening for HTN and high blood sugar. Among the participants (N = 274256) at high risk of metabolic syndrome (either having DM/HTN/ high waist/BMI$\geq$25), the weighted prevalence of DM-HTN-high waist circumference was 2.00% (n = 4644, 95% CI: 1.91 to 2.09) including 0.93% (n = 2070, 95% CI: 0.87 to 0.99) newly diagnosed cases. The weighted prevalence of Metabolic syndrome as per NCEP ATPIII criteria in the participants was 1.13% (n = 6951, 95% CI: 1.08 to 1.17).

Table 1 reports the socio-demographic and lifestyle characteristics among the participants having DM-HTN comorbidity with high waist circumference. Among the previously diagnosed cases, a majority of them were middle-aged (41–49 years) (59.12%) and females (96.18%).

Among the previously diagnosed cases of DM-HTN comorbidity with high waist circumference, 46.16% (95% CI: 43.02 to 49.32) cases were on treatment with both anti-diabetes and

**Table 1. Socio-demographic and lifestyle characteristics of DM-HTN comorbid individuals with high waist circumference.**

| Characteristics | DM-HTN comorbid with High waist circumference | | |
| --- | --- | --- | --- |
| | Previously diagnosed | Newly diagnosed | Total |
| | n (weighted %) | n (weighted %) | n (weighted %) |
| | (n = 2574) | (n = 2070) | (N = 4644) |
| **Age (years)** | | | |
| 15–30 | 291 (10.09) | 137 (6.427) | 428 (8.39) |
| 31–40 | 761 (28.66) | 689 (34.51) | 1450 (31.37) |
| 41–49 | 1522 (61.25) | 1244 (59.06) | 2766 (60.24) |
| **Sex** | | | |
| Male | 95 (3.99) | 151 (8.252) | 246 (5.965) |
| Female | 2479 (96.01) | 1919 (91.75) | 4398 (94.04) |
| **Education** | | | |
| No education | 555 (21.7) | 524 (23.28) | 1079 (22.43) |
| Primary | 370 (13.78) | 350 (19.28) | 720 (16.33) |
| Secondary | 1251 (49.65) | 982 (47.18) | 2233 (48.51) |
| Higher | 398 (14.88) | 214 (10.26) | 612 (12.74) |
| **Occupation** | | | |
| With high physical activity | 176 (40.29) | 201 (48.2) | 377 (44.41) |
| With low physical activity | 307 (59.71) | 239 (51.8) | 546 (55.59) |
| **Place of Residence** | | | |
| Rural | 1458 (50.46) | 1152 (49.08) | 2610 (49.82) |
| Urban | 1116 (49.54) | 918 (50.92) | 2034 (50.18) |
| **Wealth Index** | | | |
| Poorest | 137 (4.453) | 121 (5.579) | 258 (4.975) |
| Poorer | 289 (9.703) | 268 (12.56) | 557 (11.03) |
| Middle | 486 (18.87) | 422 (19.76) | 908 (19.28) |
| Richer | 731 (28.73) | 579 (26.68) | 1310 (27.78) |
| Richest | 931 (38.25) | 680 (35.41) | 1611 (36.94) |
| **BMI (kg/m$^2$)** | | | |
| Underweight/ Normal | 477 (15.62) | 282 (14.86) | 759 (15.27) |
| Overweight | 1053 (41.27) | 888 (40.36) | 1941 (40.85) |
| Obese | 1039 (43.11) | 896 (44.77) | 1935 (43.88) |
| **Smoking Status** | | | |
| Yes | 155 (4.294) | 176 (8.186) | 331 (6.098) |
| No | 2419 (95.71) | 1894 (91.81) | 4313 (93.9) |
| **Alcohol consumption** | | | |
| Yes | 69 (1.342) | 82 (3.836) | 151 (2.497) |
| No | 2505 (98.66) | 1988 (96.16) | 4493 (97.5) |
| **Media Exposure** | | | |
| Yes | 2285 (89.45) | 1805 (87.48) | 4090 (88.54) |
| No | 289 (10.55) | 265 (12.52) | 554 (11.46) |
| **Diet** | | | |
| High fat | 561 (21.45) | 388 (17.75) | 949 (19.74) |
| Low fat | 2013 (78.55) | 1682 (82.25) | 3695 (80.26) |
| **Additional Comorbidities** | | | |
| None | 2129 (81.33) | 1812 (87.8) | 3941 (84.33) |
| 1 or more | 445 (18.67) | 258 (12.2) | 703 (15.67) |

antihypertensive medication (full treatment), 19.14% (95% CI: 17.12 to 21.33) cases were on either anti-diabetes or antihypertensive medication but not both (partial treatment), while 34.71% (95% CI: 31.70 to 37.84) cases were on neither treatment (untreated). Table 2 reports the treatment seeking behaviour of previously diagnosed DM-HTN comorbid patients having high waist circumference. Upon crude multinomial regression analysis, males and increasing age were significantly associated with availing partial or full treatment for DM-HTN comorbidity. Similarly, higher wealth index, urban residency, increasing BMI, exposure to media and presence of additional comorbidities were positively associated with availing full treatment as compared to availing no treatment. Adjusted analysis showed that individuals belonging to age group 41–49 years (aRRR = 3.19, 95% CI: 1.96 to 5.20) and richest wealth index (aRRR = 2.81, 95% CI: 1.36 to 5.81) were more likely to be on partial treatment rather than not taking any treatment. Similarly, individuals having age 41–49 years (aRRR = 18.03, 95% CI: 9.87 to 32.94), richest wealth index (aRRR = 2.42, 95% CI: 1.14 to 5.11), urban residency (aRRR = 1.62, 95% CI: 1.15 to 2.27), obesity (aRRR = 1.56, 95% CI: 1.02 to 2.37), media exposure (aRRR = 1.52, 95% CI: 1.03 to 2.55) and additional comorbidities (aRRR = 1.60, 95% CI: 1.16 to 2.22) were more likely to be on full treatment as compared to not being on either treatment. Further, females (aRRR = 0.42, 95% CI: 0.20 to 0.87) were less likely to be on full treatment than no treatment as compared to males.

Table 3 reports the association of the socio-demographic and lifestyle characteristics among the cases of DM-HTN comorbidity with high-waist circumference stratified by their time of diagnosis (previous, new, total). On adjusted analysis, increasing age, female sex, higher wealth index, increasing BMI and exposure to media had significantly higher rate ratios (RR) of having a previous diagnosis of DM-HTN comorbidity with high-waist circumference. Similarly, increasing age, lower education levels, urban residence, higher wealth index and high BMI were the predictors associated with newly diagnosed cases of DM-HTN comorbidity with high-waist circumference. The risk factors that were significantly associated with occurrence of DM-HTN-comorbidity with high waist circumference in the total cases (both previously diagnosed and new cases) were the middle age group (41–49 years) compared to younger aged cases (aRR = 6.12, 95% CI: 5.24 to 7.15), females compared to male cases (aRR = 1.62, 95% CI: 1.25 to 2.10), higher compared to no education (aRR = 0.84, 95% CI: 0.71 to 0.99), cases belonging to richest wealth index compared to lower wealth quintiles (aRR = 1.91, 95% CI: 1.52 to 2.40), obesity (aRR = 4.68, 95% CI: 4.06 to 5.40), high fat diet compared to low fat diet (aRR = 1.18, 95% CI: 1.06 to 1.30) and presence of additional comorbidities compared to no other self-reported comorbidities (aRR = 1.47, 95% CI: 1.32 to 1.63).

Furthermore, less than half (46.83%) of the cases with DM-HTN comorbidity with high waist circumference had both controlled BP (<140/90) and blood glucose levels (RBS<180). However, nearly 18% of the cases had both suboptimal blood pressure and glycemic control (Fig 1). Significant regional variation was observed in the prevalence of DM-HTN comorbidity with high waist circumference across Indian states and union territories with Chandigarh having the highest prevalence (2.53%), followed by Tamil Nadu (2.31%), while the lowest was observed in Meghalaya (0.12%) (Fig 2).

## 4. Discussion

The present study observed the prevalence of DM-HTN comorbidity with high waist circumference equivalent to metabolic syndrome (Metabolic syndrome) was lower (0.75%) compared to regional estimates of metabolic syndrome [31, 32] although it is an underestimation as the case definitions in this study did not account for elevated triglyceride or HDL-C levels which were unavailable, and also the lack of older adults and elderly in the sample. Further, the

**Table 2. Treatment-seeking behaviour among previously diagnosed DM-HTN comorbid patients with high waist circumference.**

| Variables | Not on Treatment n (weighted %) (n = 1026) | Partial Treatment (n = 501) | | Full treatment n (weighted %) (n = 1047) | Crude RRR[1] (95% CI) | Crude RRR[2] (95% CI) | aRRR[1] (95% CI) | aRRR[2] (95% CI) |
|---|---|---|---|---|---|---|---|---|
| | | On anti-diabetes medication only (n = 298) n (weighted %) | On anti-hypertensive medication only (n = 203) n (weighted %) | | | | | |
| **Age (years)** | | | | | | | | |
| 15–30 | 217 (75.17) | 24 (8.28) | 25 (7.83) | 25 (8.718) | Ref | Ref | Ref | Ref |
| 31–40 | 407 (49.62) | 83 (12.54) | 56 (6.51) | 215 (31.34) | 1.79 [1.08, 2.97] * | 5.45 [3.00, 9.88] ** | 1.42 [0.86, 2.35] | 4.16 [2.25, 7.69] ** |
| 41–49 | 402 (21.06) | 191 (12.75) | 122 (6.93) | 807 (59.26) | 4.36 [2.69, 7.07] ** | 24.26 [13.71,42.91] ** | 3.19 [1.96, 5.20] ** | 18.03 [9.87, 32.94] ** |
| **Sex** | | | | | | | | |
| Male | 31 (14.23) | 12 (22.48) | 7 (7.63) | 45 (55.67) | Ref | Ref | Ref | Ref |
| Female | 995 (35.56) | 286 (11.81) | 196 (6.87) | 1002 (45.76) | 0.25 [0.09, 0.68] * | 0.33 [0.16, 0.68] * | 0.26 [0.10, 0.66] | 0.42 [0.20, 0.87] * |
| **Education** | | | | | | | | |
| No education | 184 (28.49) | 68 (12.37) | 59 (10.20) | 244 (48.93) | Ref | Ref | Ref | Ref |
| Primary | 123 (30.08) | 47 (12.59) | 30 (7.08) | 170 (50.25) | 0.82 [0.51, 1.34] | 0.97 [0.65, 1.46] | 0.79 [0.48, 1.30] | 0.93 [0.59, 1.44] |
| Secondary | 507 (33.73) | 145 (13.08) | 85 (5.18) | 514 (48.01) | 0.68 [0.46, 1.02] | 0.83 [0.60, 1.15] | 0.60 [0.38, 0.94] * | 0.75 [0.52, 1.10] |
| Higher | 212 (51.31) | 38 (8.93) | 29 (7.64) | 119 (32.12) | 0.41 [0.25, 0.67] ** | 0.36 [0.24, 0.55] ** | 0.33 [0.19, 0.58] ** | 0.32 [0.20, 0.53] ** |
| **Occupation** | | | | | | | | |
| With higher physical activity | 70 (28.14) | 21 (21.08) | 14 (7.66) | 71 (43.12) | Ref | Ref | - | - |
| With lower physical activity | 103 (28.63) | 36 (12.78) | 30 (10.80) | 138 (47.78) | 0.81 [0.38,1.69] | 1.09 [0.57, 2.08] | | |
| **Place of Residence** | | | | | | | | |
| Rural | 644 (40.82) | 172 (12.03) | 127 (7.81) | 515 (39.34) | Ref | Ref | Ref | Ref |
| Urban | 382 (28.48) | 126 (12.45) | 76 (5.97) | 532 (53.1) | 1.33 [0.96, 1.85] | 1.93 [1.43, 2.62] ** | 1.09 [0.75, 1.58] | 1.62 [1.15, 2.27] * |
| **Wealth Index** | | | | | | | | |
| Poorest | 73 (56.21) | 14 (9.09) | 11 (6.42) | 39 (28.28) | Ref | Ref | Ref | Ref |
| Poorer | 149 (44.47) | 27 (11.43) | 22 (6.94) | 91 (37.16) | 1.50 [0.75, 3.00] | 1.66 [0.88, 3.12] | 1.58 [0.75, 3.30] | 1.60 [0.75, 3.44] |
| Middle | 211 (40.8) | 58 (10.89) | 33 (4.75) | 184 (43.56) | 1.39 [0.72, 2.66] | 2.12 [1.17, 3.84] * | 1.52 [0.76, 3.06] | 1.81 [0.87, 3.74] |
| Richer | 269 (31.07) | 95 (13.88) | 70 (8.51) | 297 (46.53) | 2.61 [1.39, 4.90] * | 2.98 [1.67, 5.31] ** | 2.97 [1.46, 6.03] * | 2.26 [1.08, 4.71] * |
| Richest | 324 (29.45) | 104 (12.24) | 67 (6.79) | 436 (51.52) | 2.34 [1.26, 4.36] * | 3.48 [1.97, 6.14] ** | 2.81 [1.36, 5.81] * | 2.42 [1.14, 5.11] * |
| **BMI (kg/m²)** | | | | | | | | |
| Underweight/ Normal | 273 (41.57) | 49 (13.64) | 31 (6.85) | 124 (37.93) | Ref | Ref | Ref | Ref |
| Overweight | 426 (39.46) | 129 (12.02) | 86 (6.63) | 412 (41.89) | 0.96 [0.63, 1.47] | 1.16 [0.81, 1.68] | 0.78 [0.50, 1.22] | 0.94 [0.62, 1.40] |
| Obese | 326 (27.77) | 120 (11.99) | 86 (7.21) | 507 (53.03) | 1.40 [0.90, 2.20] | 2.09 [1.43, 3.06] ** | 1.07 [0.67, 1.70] | 1.56 [1.02, 2.37] * |
| **Smoking Status** | | | | | | | | |

(*Continued*)

**Table 2.** (Continued)

| Variables | Not on Treatment n (weighted %) (n = 1026) | Partial Treatment (n = 501) | | Full treatment n (weighted %) (n = 1047) | Crude RRR[1] (95% CI) | Crude RRR[2] (95% CI) | aRRR[1] (95% CI) | aRRR[2] (95% CI) |
|---|---|---|---|---|---|---|---|---|
| | | On anti-diabetes medication only (n = 298) n (weighted %) | On anti-hypertensive medication only (n = 203) n (weighted %) | | | | | |
| Yes | 51 (29.5) | 22 (25.52) | 15 (4.69) | 67 (40.29) | 1.92 [0.82, 4.47] | 1.03 [0.59, 1.80] | | |
| No | 975 (34.94) | 276 (11.64) | 188 (7.00) | 980 (46.42) | Ref | Ref | - | - |
| **Alcohol consumption** | | | | | | | | |
| Yes | 24 (23.4) | 6 (13.60) | 5 (6.78) | 34 (56.23) | 1.59 [0.54, 4.68] | 1.82 [0.72, 4.62] | - | - |
| No | 1002 (34.86) | 292 (12.22) | 198 (6.90) | 1013 (46.02) | Ref | Ref | | |
| **Media Exposure** | | | | | | | | |
| Yes | 896 (33.53) | 267 (12.31) | 173 (6.71) | 949 (47.45) | 1.26 [0.81, 1.96] | 1.80 [1.22, 2.64] * | 1.17 [0.70, 1.96] | 1.62 [1.03, 2.55] * |
| No | 130 (44.67) | 31 (11.68) | 30 (8.45) | 98 (35.2) | Ref | Ref | Ref | Ref |
| **Diet** | | | | | | | | |
| High fat | 217 (32.9) | 60 (13.03) | 44 (6.53) | 240 (47.54) | 1.10 [0.75, 1.61] | 1.11 [0.81, 1.52] | | |
| Low fat | 809 (35.2) | 238 (12.02) | 159 (7.00) | 807 (45.78) | Ref | Ref | - | - |
| **Comorbidities** | | | | | | | | |
| None | 897 (36.84) | 250 (12.43) | 166 (6.62) | 816 (44.12) | Ref | Ref | Ref | Ref |
| 1 or more | 129 (25.43) | 48 (11.43) | 37 (8.12) | 231 (55.02) | 1.49 [1.003, 2.20] * | 1.81 [1.33, 2.45] ** | 1.46 [0.98, 2.18] | 1.60 [1.16, 2.22] * |

Abbreviations: RRR, Relative Risk Ratio; aRRR, Adjusted Relative Risk Ratio; CI, Confidence Interval, Ref, Reference Category; BMI, Body Mass Index

Variables found to be significant in the crude multinomial regression were included in the adjusted model

*Crude RRR[1] = Partial Treatment versus No Treatment; Crude RRR[2] = Full Treatment versus No Treatment*

*aRRR[1] = Partial Treatment versus No Treatment; aRRR[2] = Full Treatment versus No Treatment;*

* P < 0.05

** P < 0.001

Model Akaike's information criterion (AIC) = 4934.86; Bayesian information criterion (BIC) = 5122.10

observed prevalence of newly diagnosed participants, although representing a small percentage statistically, holds significant clinical implications within the context of India's healthcare landscape. Undiagnosed cases of metabolic syndrome contribute to a substantial burden of preventable cardiovascular complications and other adverse health outcomes [33]. Given the potential economic burden and public health impact associated with untreated or underdiagnosed metabolic syndrome, early detection and intervention strategies are imperative.

The overall prevalence of DM-HTN comorbidity with high waist circumference was significantly higher in females, a finding consistent with the findings of a meta-analysis of studies from India [11]. The presence of Metabolic syndrome in this study was also associated with increasing age and higher BMI, findings that are consistent with the global evidence [34–36]. Furthermore, in this study, people living in urban areas were 1.2 times more likely to have Metabolic syndrome as compared to rural residents suggestive of the linkage of adverse social determinants such as sedentary lifestyle, dietary changes and also stress in urban areas [34].

The present study findings indicate that higher education levels decrease the risk of having undiagnosed metabolic syndrome in accordance with a study conducted in China [37],

Table 3. Factors associated with DM-HTN-abdominal obesity among high-risk young and middle-aged individuals[a].

| Variables | Previously diagnosed DM-HTN-High waist | | Newly diagnosed DM-HTN-High waist | | Total DM-HTN-High waist | |
|---|---|---|---|---|---|---|
| | (n = 2574) | | (n = 2070) | | (n = 4644) | |
| | Crude RR (95% CI) | aRR (95% CI) | Crude RR (95% CI) | aRR (95% CI) | Crude RR (95% CI) | aRR (95% CI) |
| **Age (years)** | | | | | | |
| 15–30 | Ref | Ref | Ref | Ref | Ref | Ref |
| 31–40 | 2.56 [2.11, 3.10] ** | 2.21 [1.40, 3.48] * | 4.91 [3.74, 6.45] ** | 4.01 [3.04, 5.29] ** | 3.37 [2.87, 3.95] ** | 2.85 [2.43, 3.35] ** |
| 41–49 | 6.18 [5.12, 7.45] ** | 7.48 [4.82, 11.62] ** | 9.50 [7.32, 12.32] ** | 7.59 [5.80, 9.95] ** | 7.30 [6.27, 8.50] ** | 6.12 [5.24, 7.15] ** |
| **Sex** | | | | | | |
| Male | Ref | Ref | Ref | - | Ref | Ref |
| Female | 2.90 [1.95, 4.30] ** | 2.07 [1.28, 3.33] * | 1.34 [0.91, 1.97] | | 1.90 [1.43, 2.52] ** | 1.62 [1.25, 2.10] ** |
| **Education** | | | | | | |
| No education | Ref | - | Ref | Ref | Ref | Ref |
| Primary | 1.21 [0.94, 1.34] | | 1.47 [1.21, 1.78] ** | 1.38 [1.14, 1.68] * | 1.29 [1.13, 1.47] ** | 1.20 [1.05, 1.37] * |
| Secondary | 1.09 [0.95, 1.24] | | 0.96 [0.82, 1.12] | 0.98 [0.83, 1.16] | 1.03 [0.93, 1.14] | 1.03 [0.92, 1.15] |
| Higher | 0.94 [0.78, 1.12] | | 0.60 [0.49, 0.75] ** | 0.65 [0.51, 0.84] * | 0.78 [0.67, 0.89] ** | 0.84 [0.71, 0.99] * |
| **Occupation** | | | | | | |
| With higher physical activity | Ref | Ref | Ref | - | Ref | - |
| With lower physical activity | 1.49 [1.10, 2.01] * | 1.004 [0.76, 1.33] | 1.08 [0.77, 1.52] | | 1.26 [0.99, 1.60] | |
| **Place of Residence** | | | | | | |
| Rural | Ref | Ref | Ref | Ref | Ref | Ref |
| Urban | 1.51 [1.32, 1.71] ** | 0.90 [0.65, 1.25] | 1.60 [1.40, 1.83] ** | 1.23 [1.05, 1.43] * | 1.54 [1.41, 1.70] ** | 1.09 [0.98, 1.21] |
| **Wealth Index** | | | | | | |
| Poorest | Ref | Ref | Ref | Ref | Ref | Ref |
| Poorer | 1.57 [1.20, 2.06] * | 1.28 [0.62, 2.66] | 1.64 [1.20, 2.23] * | 1.39 [1.03, 1.89] * | 1.60 [1.30, 1.97] ** | 1.36 [1.11, 1.68] * |
| Middle | 2.40 [1.85, 3.10] ** | 1.70 [0.84, 3.44] | 2.03 [1.52, 2.72] ** | 1.46 [1.09, 1.95] * | 2.19 [1.80, 2.67] ** | 1.59 [1.29, 1.95] ** |
| Richer | 3.10 [2.42, 3.97] ** | 2.07 [1.04, 4.12] * | 2.34 [1.78, 3.08] ** | 1.50 [1.12, 2.01] * | 2.68 [2.22, 3.23] ** | 1.74 [1.42, 2.14] ** |
| Richest | 3.82 [2.99, 4.88] ** | 2.94 [1.43, 6.04] * | 2.88 [2.17, 3.82] ** | 1.68 [1.22, 2.33] * | 3.30 [2.73, 3.99] ** | 1.91 [1.52, 2.40] ** |
| **BMI (kg/m$^2$)** | | | | | | |
| Underweight/ Normal | Ref | Ref | Ref | Ref | Ref | Ref |
| Overweight | 1.95 [1.66, 2.30] ** | 1.65 [1.08, 2.54] * | 2.07 [1.68, 2.55] ** | 1.90 [1.53, 2.36] | 1.98 [1.73, 2.25] ** | 1.80 [1.57, 2.05] ** |
| Obese | 5.89 [4.97, 6.98] ** | 5.31 [3.41, 8.24] ** | 6.63 [5.34, 8.23] ** | 5.33 [4.24, 6.72] ** | 6.13 [5.36, 7.02] ** | 4.68 [4.06, 5.40] ** |
| **Smoking Status** | | | | | | |
| Yes | 0.52 [0.39, 0.69] ** | 1.27 [0.63, 2.56] | 1.03 [0.71, 1.50] | - | 0.75 [0.57, 0.99] * | 0.996 [0.77, 1.28] |
| No | Ref | Ref | Ref | | Ref | Ref |
| **Alcohol consumption** | | | | | | |
| Yes | 0.36 [0.24, 0.53] ** | 0.74 [0.37, 1.49] | 1.06 [0.57, 1.98] | - | 0.68 [0.43, 1.08] | |
| No | Ref | Ref | Ref | | Ref | - |
| **Media Exposure** | | | | | | |
| Yes | 1.57 [1.33, 1.85] ** | 2.94 [1.52, 5.67] * | 1.31 [1.09, 1.56] * | Ref | 1.43 [1.27, 1.61] ** | 1.06 [0.93, 1.21] |
| No | Ref | Ref | Ref | 0.99 [0.82, 1.22] | Ref | Ref |
| **Diet** | | | | | | |

*(Continued)*

**Table 3.** (Continued)

| Variables | Previously diagnosed DM-HTN-High waist | | Newly diagnosed DM-HTN-High waist | | Total DM-HTN-High waist | |
|---|---|---|---|---|---|---|
| | (n = 2574) | | (n = 2070) | | (n = 4644) | |
| | Crude RR (95% CI) | aRR (95% CI) | Crude RR (95% CI) | aRR (95% CI) | Crude RR (95% CI) | aRR (95% CI) |
| High fat | 1.34 [1.16, 1.54] ** | 0.83 [0.59, 1.16] | 1.06 [0.91, 1.24] | | 1.20 [1.08, 1.34] ** | 1.18 [1.06, 1.30] * |
| Low fat | Ref | Ref | Ref | - | Ref | Ref |
| **Additional Comorbidities** | | | | | | |
| None | Ref | Ref | Ref | Ref | Ref | Ref |
| 1 or more | 2.73 [2.38, 3.12] ** | 1.24 [0.89, 1.72] | 1.66 [1.39, 1.98] ** | 1.12 [0.94, 1.34] | 2.21 [1.98, 2.46] ** | 1.47 [1.32, 1.63] ** |
| **Model Fitness** | | | | | | |
| **Akaike's information criterion (AIC)** | - | 5042.91 | - | 22056.37 | - | 42396.31 |
| **Bayesian information criterion (BIC)** | - | 5197.69 | - | 22214.17 | - | 42585.46 |

Abbreviations: RR, Rate Ratio; aRR, Adjusted Rate Ratio; CI, Confidence Interval; Ref, Reference Category; DM, Diabetes Mellitus; HTN, Hypertension; BMI, Body Mass Index

Variables found to be significant in the crude Poisson regression were included in the respective adjusted models

* P < 0.05

** P < 0.001

[a]Denominator (N = 274256) taken as individuals with higher-risk of metabolic syndrome, i.e., detected with DM or HTN or high waist circumference or high BMI (≥25.0)

suggestive of the impact of adverse social determinants of health contributed to delayed screening and diagnosis of DM and HTN in individuals with lower educational status compared with their counterparts. Less than half of the cases in our sample (46.8%) having DM-HTN comorbidity with high waist circumference and previously initiated on treatment had poorly controlled blood pressure and blood glucose levels that may reflect either poor medical adherence, or poor response to hypertension treatment [38] and insulin resistance [39].

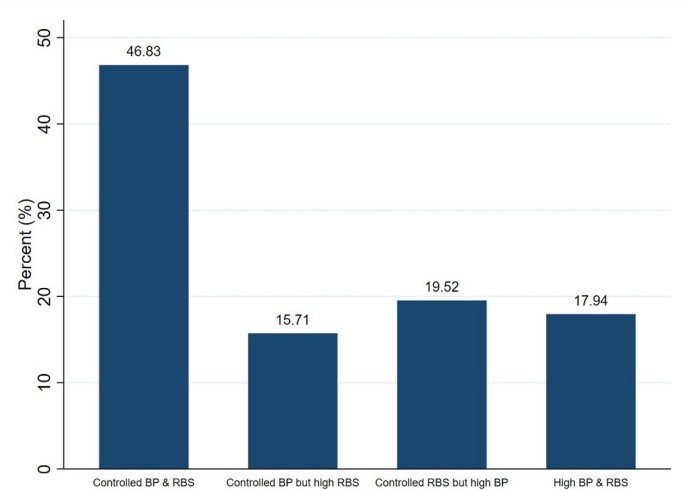

**Fig 1. Health outcomes of participants with DM-HTN comorbidity along with high waist circumference (both old and new cases, N = 2520).**

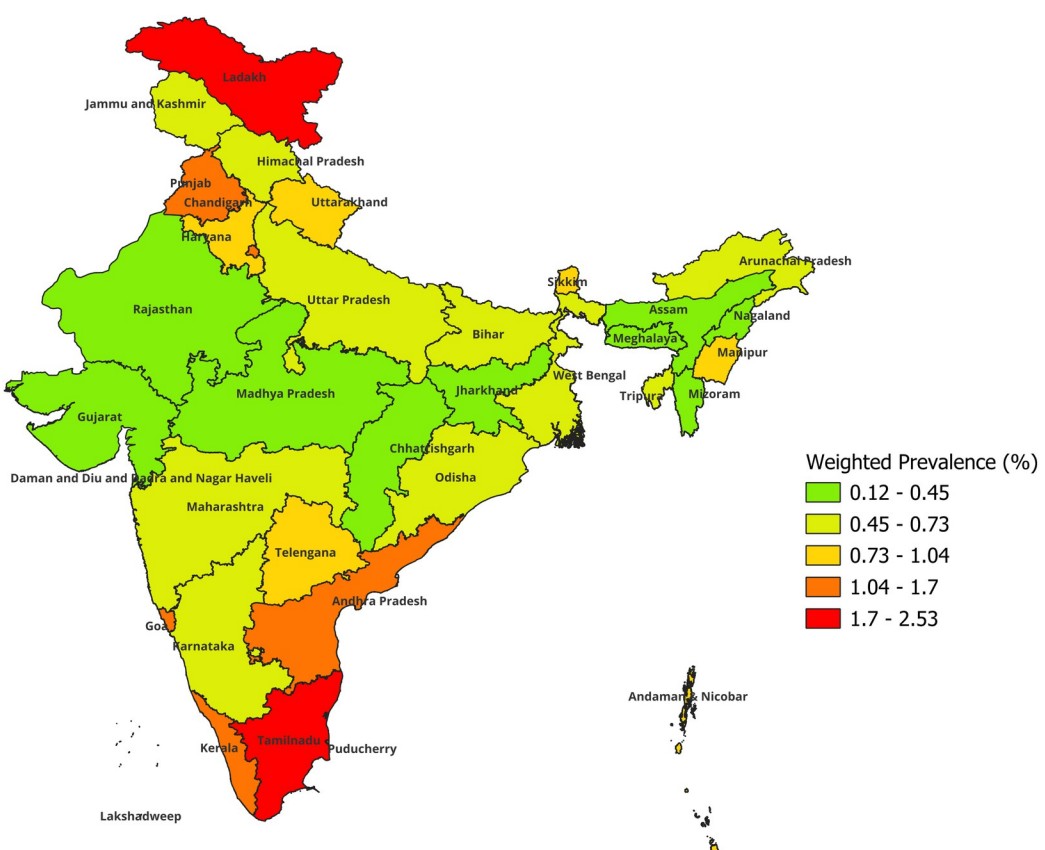

**Fig 2. Prevalence of DM-HTN-abdominal obesity in high-risk young and middle-aged persons in India.** This map used is the authors' own creation. For the base layer of India, we utilized the free GIS file available from https://spatialdata. dhsprogram.com/boundaries/#view=table&countryId=IA.

Furthermore, in this study, the magnitude of metabolic syndrome like phenomenon in the 15–49 age-group indicated significant variation across states and union territories of India. The Southern Indian states had comparatively higher prevalence which could be associated with both improved health system functioning translating into increased screening and diagnosis, greater awareness due to higher literacy rates in the population, as also sociocultural practices especially dietary behaviors contributing to the phenomenon [40].

Furthermore, patients with Metabolic syndrome satisfying criteria of having high waist circumference, elevated blood pressure and triglyceride levels (S1 Table) require early and effective treatment to prevent the onset and progression of cardiovascular complication [41]. However, in this study, nearly one in two cases with Metabolic syndrome were undiagnosed that were detected on survey-based screening of blood pressure and blood glucose levels signifying an iceberg phenomenon of disease wherein only a fraction of the total cases is clinically diagnosed and initiated on effective treatment. Current evidence suggests the efficacy of early management of metabolic syndrome components especially prior to occurrence of any cardiovascular event is vital for reducing cardiovascular risk [42, 43]. Despite the limitations of 10-year cardiovascular risk-based approaches in the young populations, treatment of hypertensive with blood pressure SBP >140 and DBP >90, and high blood glucose in those with diabetes is mandated by most standard guidelines [44, 45].

Consequently, public health strategies to control the pandemic of Metabolic syndrome driven cardiovascular disease burden in the developing world including India requires scaling up of screening individuals above age 30 especially those with any high-risk factor for Metabolic syndrome for HTN, DM, abdominal obesity, and elevated blood triglyceride levels. Furthermore, initiation of effective treatment for cases diagnosis with HTN, DM, or Metabolic syndrome is of paramount significance. In this context, the revamped national program for non-communicable diseases (NP-NCD 2023–2030) in India has set an ambitious 75:25 initiative for placing 75 million patients with HTN or DM on standard by 2025 [46].

### 4.1 Strengths and limitations of the study

The study has certain strengths including its analysis of a large sample size and nationally representative dataset with adequate geographical diversity that supports generalizability of the findings. However, there are also some major study limitations. First, the cross-sectional design of the study does not permit assessment of temporal association and precludes causal assessment. Second, information for two important components of metabolic syndrome, serum cholesterol and triglyceride levels were not available in the NFHS datasets that is likely to have contributed to significant underestimation of the burden of the problem. Third, there may be some recall bias, although blood pressure and RBS measurements were conducted as part of the study. However, the lack of fasting blood glucose and glycated haemoglobin measurements precluded the validated estimation of glycemic control in the patients. Fourth, it is important to note that the survey lacks information regarding the time of diagnosis for HTN or DM, limiting the ability to assess the impact of the duration of these conditions on the observed associations. Lastly, no direct information on physical activity was available in the datasets so a surrogate was obtained from the individual's occupation which may not accurately reflect the physical activity status of the participants.

### 4.2 Conclusion and recommendations

In conclusion, more than half of young and middle aged-population in India with DM-HTN-abdominal obesity triad are not initiated on treatment for DM and HTN comorbidities especially those from socioeconomically disadvantaged groups, while a majority of the previously diagnosed cases have uncontrolled blood pressure and poor glycemic control. Females had both significantly higher risk of occurrence of the condition with accompanying adverse treatment seeking behaviour, while those of richest wealth index had higher risk of occurrence with improved treatment seeking behaviour compared to individuals belonging to lower wealth index.

The poor cascade of care for DM and HTN in these high-risk group of patients may substantially increase their risk for early progression and severity of microvascular and macrovascular complications especially cardiovascular disease. However, further research on treatment strategies for reducing the risk specific cardiovascular outcomes within this younger demographic is urgently warranted.

Early diagnosis of cases of metabolic syndrome including DM and HTN, with early initiation of effective medication and treatment, regular monitoring and targeted reduction of blood pressure, blood glucose, and body weight through primary health system strengthening and community engagement remain the cornerstone of reducing complications from these debilitating health conditions.

## Supporting information

**S1 Table. Diagnostic criteria of Metabolic syndrome as per NCEP ATP-III [10].**
(DOCX)

**S2 Table. Descriptive statistics for continuous variables.**
(DOCX)

## Acknowledgments

The authors thank DHS for providing the NFHS-5 datasets.

## Author Contributions

**Conceptualization:** Saurav Basu.

**Formal analysis:** Vansh Maheshwari, Mansi Malik.

**Methodology:** Saurav Basu, Vansh Maheshwari, Mansi Malik, Kara Barzangi, Refaat Hassan.

**Visualization:** Saurav Basu, Refaat Hassan.

**Writing – original draft:** Saurav Basu, Vansh Maheshwari, Kara Barzangi.

**Writing – review & editing:** Saurav Basu, Vansh Maheshwari, Mansi Malik, Kara Barzangi, Refaat Hassan.

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
