## [Decision Letter · Decision Letter 0]

19 Oct 2023

PGPH-D-23-01679

The burden and care cascade in young and middle-aged patients with Diabetes hypertension comorbidity with abdominal obesity in India: a nationally representative cross-sectional survey

Dear Dr. Barzangi,

Thank you for submitting your manuscript to PLOS Global Public Health. After careful consideration, we feel that it has merit but does not fully meet PLOS Global Public Health’s publication criteria as it currently stands. Therefore, we invite you to submit a revised version of the manuscript that addresses the points raised during the review process.

We look forward to receiving your revised manuscript.

Kind regards,

Rajat Das Gupta, M.D.

Academic Editor

Journal Requirements:

1. Tables should not be uploaded as individual files. Please remove these files and include the Tables in your manuscript file as editable, cell-based objects. For more information about how to format tables, see our guidelines:

https://journals.plos.org/globalpublichealth/s/tables

2. We have noticed that you have uploaded Supporting Information files, but you have not included a list of legends. Please add a full list of legends for your Supporting Information files after the references list. 

3. Please amend your Data Availability Statement and indicate where the data may be found.

4. Some material included in your submission may be copyrighted. According to PLOS’s copyright policy, authors who use figures or other material (e.g., graphics, clipart, maps) from another author or copyright holder must demonstrate or obtain permission to publish this material under the Creative Commons Attribution 4.0 International (CC BY 4.0) License used by PLOS journals. Please closely review the details of PLOS’s copyright requirements here: PLOS Licenses and Copyright. If you need to request permissions from a copyright holder, you may use PLOS's Copyright Content Permission form.

Potential Copyright Issues:

Fig 2: please (a) provide a direct link to the base layer of the map (i.e., the country or region border shape) and ensure this is also included in the figure legend; and (b) provide a link to the terms of use / license information for the base layer image or shapefile. We cannot publish proprietary or copyrighted maps (e.g. Google Maps, Mapquest) and the terms of use for your map base layer must be compatible with our CC-BY 4.0 license. 

"

Additional Editor Comments (if provided):

Reviewers' comments:

Reviewer's Responses to Questions

**Comments to the Author**

1. Does this manuscript meet PLOS Global Public Health’s publication criteria? Is the manuscript technically sound, and do the data support the conclusions? The manuscript must describe methodologically and ethically rigorous research with conclusions that are appropriately drawn based on the data presented.

Reviewer #1: Yes

2. Has the statistical analysis been performed appropriately and rigorously?

Reviewer #1: No

3. Have the authors made all data underlying the findings in their manuscript fully available (please refer to the Data Availability Statement at the start of the manuscript PDF file)?

Reviewer #1: No

4. Is the manuscript presented in an intelligible fashion and written in standard English?

Reviewer #1: Yes

5. Review Comments to the Author

Reviewer #1: 1. Valuable topic. A number of points need to be highlighted.

2. better to highlight only diabetes, hypertension, metabolic syndrome and India as keywords.

3. can you re-arrange the introduction as follow:

a. brief introduction or definition of the health problem in your topic.

b. directly give the prevalence and incidence globally and in the research area(if there is available data).

c. discuss the definition of DM,HT, metabolic syndrome and abdominal obesity as series of diseases in un separated idea.

4. is this the exactly age(18-49)?

5. mention the instruction of data collection

6. define the standard level of the measured variable.

7. questionnaire date are represented as inclusion, exclusion criteria or instruction information to collect or to choose sample.

8. Re-strained your methods to include Study design and area, Study subjects, Inclusion criteria, Exclusion criteria, methods for data Collection and Preparation of Samples.

9. set your participants in methodology as groups i.e; diabetic hypertensive, diabetic hypertension with abdomen obesity or with metabolic syndrome.

10. criteria of disease+criteria of participant. you must set it for one time in inclusion and inclusion criteria of participants then you can write the standard criteria of your collecting your sample.

11. write about the methods of analysis. you can describe the method in the text of table of results briefly.

12. mean SD is not recorded. please add or insert table that include the average mean of variables.

13. can the author determine the time of affecting by DM and H, or what was the firstly diagnosed, DM Or HT.

14. please point to the table which indicate the criteria of metabolic syndrome.

15. Divide your discussion into sub-title include :

- conclusion

- recommendations

- limitation of the study

- A acknowledgements.

6. PLOS authors have the option to publish the peer review history of their article (what does this mean?). If published, this will include your full peer review and any attached files.

**Do you want your identity to be public for this peer review?** For information about this choice, including consent withdrawal, please see our Privacy Policy.

Reviewer #1: **Yes: **Nahla Ahmed Mohammed Abderhman

---

## [Decision Letter · Decision Letter 1]

27 Feb 2024

PGPH-D-23-01679R1

The burden and care cascade in young and middle-aged patients with Diabetes hypertension comorbidity with abdominal obesity in India: a nationally representative cross-sectional survey

Dear Dr. Barzangi,

Thank you for submitting your manuscript to PLOS Global Public Health. After careful consideration, we feel that it has merit but does not fully meet PLOS Global Public Health’s publication criteria as it currently stands. Therefore, we invite you to submit a revised version of the manuscript that addresses the points raised during the review process.

We look forward to receiving your revised manuscript.

Kind regards,

Rajat Das Gupta, M.D.

Academic Editor

Journal Requirements:

Additional Editor Comments (if provided):

Reviewers' comments:

Reviewer's Responses to Questions

**Comments to the Author**

1. If the authors have adequately addressed your comments raised in a previous round of review and you feel that this manuscript is now acceptable for publication, you may indicate that here to bypass the “Comments to the Author” section, enter your conflict of interest statement in the “Confidential to Editor” section, and submit your "Accept" recommendation.

Reviewer #1: All comments have been addressed

Reviewer #2: (No Response)

2. Does this manuscript meet PLOS Global Public Health’s publication criteria? Is the manuscript technically sound, and do the data support the conclusions? The manuscript must describe methodologically and ethically rigorous research with conclusions that are appropriately drawn based on the data presented.

Reviewer #1: Yes

Reviewer #2: Partly

3. Has the statistical analysis been performed appropriately and rigorously?

Reviewer #1: Yes

Reviewer #2: Yes

4. Have the authors made all data underlying the findings in their manuscript fully available (please refer to the Data Availability Statement at the start of the manuscript PDF file)?

Reviewer #1: Yes

Reviewer #2: Yes

5. Is the manuscript presented in an intelligible fashion and written in standard English?

Reviewer #1: Yes

Reviewer #2: Yes

6. Review Comments to the Author

Reviewer #1: 1. thank you much for your response

study design and data source:

2. Add the study design (cross sectional study)

3. reference is enough. author can delete the highlighted sentense.

4. better to start the paragraph by those points(re-arrange points).

5. better to classify the study participants in a table instead of text, according to specific criteria, for Diabetes mellitus, Hypertension and dyslipidemia. then add the source of classification below the table.

6. add the table and their source instead.

7. all questionnaire date which obtained from the respondent were included in the analysis. no need to duplicate information in different way.

Reviewer #2: Congratulations on your efforts to analyse a very big data set with a specific research question that was adequately answered from your analysis.

Please address the following comments that will strengthen the manuscript and enhance clarity:

1. Methodology:

i) Was there a pre-study training exercise for the data collection research team? This need to be mentioned to give.

assurance that there was standardization of the data collected and guarantee reproducibility.

ii) Age was such an important outcome of the survey. Was it self-reported of objectively determined using official

government documents like a passport, Birth Certificate, Driving licence, and related documents?

iii)Outside obtaining written consent to participate in the study please clarify if further consent to take waist

circumference: this is only mentioned for BP and sugar measurements. This is important to ensure that the

highest ethical standards were employed in the conduct of the study.

iv) The initial definition of DM is limiting (2nd sentence on page 10). though subsequent section in the manuscript

clarify this further (correct definition). Please correct this to ensure consistency.

v) The observation that there were regional variations in the observed primary outcome of the study is hanging.

What does this exactly mean?

vi) There is silence on how women who could have been pregnant were handled during the data collection. The female population studied were all in the bracket of childbearing potential. Was pregnancy an exclusion criterion? If so was it self reported on based on some documentation?

2. Discussion section:

i)The number of undiagnosed participants may sound statistically a small percentage; but clinically significant. This

need to come out because it is important as already captured in your recommendation.

ii)The discussion point that more than half of those with the primary outcome were not on treatment need to be

backed with evidence from literature that justifies if treatment interventions of this specific fairlyl young

population yields positive cardiovascular outcomes (CV mortality, strokes, Heart Failure, Coronary events).

Conclusion

i) This is not strictly based on the data presented. First, the conclusion statement should be based on your study

objectives/ questions namely: ascertain burden, determinants and care cascade as related to the primary

outcome of the study. The conclusion statement starts with the last aspect; but again it has a factual error. Your

data say that 19.4% of the total participants with the primary outcome of interest with prior diagnosis (of DM,

Hypertension) were already on full treatment; while 34.715 were on partial treatment. The total is 53.85%;

much ore than the stated figure in the conclusion. Additionally, those with undiagnosed disease could not have

been on treatment for obvious reason.

ii) Thhe recommendation needs to be backed with more discussion as captured above (2 ii).

7. PLOS authors have the option to publish the peer review history of their article (what does this mean?). If published, this will include your full peer review and any attached files.

**Do you want your identity to be public for this peer review?** For information about this choice, including consent withdrawal, please see our Privacy Policy.

Reviewer #1: **Yes: **Nahla Ahmed Mohammed Abderhman

Reviewer #2: No

---

## [Decision Letter · Decision Letter 2]

12 Apr 2024

PGPH-D-23-01679R2

The burden and care cascade in young and middle-aged patients with Diabetes hypertension comorbidity with abdominal obesity in India: a nationally representative cross-sectional survey

Dear Dr. Barzangi,

Thank you for submitting your manuscript to PLOS Global Public Health. After careful consideration, we feel that it has merit but does not fully meet PLOS Global Public Health’s publication criteria as it currently stands. Therefore, we invite you to submit a revised version of the manuscript that addresses the points raised during the review process.

We look forward to receiving your revised manuscript.

Kind regards,

Rajat Das Gupta, M.D.

Academic Editor

Journal Requirements:

Additional Editor Comments (if provided):

Reviewers' comments:

Reviewer's Responses to Questions

**Comments to the Author**

1. If the authors have adequately addressed your comments raised in a previous round of review and you feel that this manuscript is now acceptable for publication, you may indicate that here to bypass the “Comments to the Author” section, enter your conflict of interest statement in the “Confidential to Editor” section, and submit your "Accept" recommendation.

Reviewer #1: All comments have been addressed

Reviewer #2: All comments have been addressed

2. Does this manuscript meet PLOS Global Public Health’s publication criteria? Is the manuscript technically sound, and do the data support the conclusions? The manuscript must describe methodologically and ethically rigorous research with conclusions that are appropriately drawn based on the data presented.

Reviewer #1: Yes

Reviewer #2: Yes

3. Has the statistical analysis been performed appropriately and rigorously?

Reviewer #1: Yes

Reviewer #2: Yes

4. Have the authors made all data underlying the findings in their manuscript fully available (please refer to the Data Availability Statement at the start of the manuscript PDF file)?

Reviewer #1: Yes

Reviewer #2: Yes

5. Is the manuscript presented in an intelligible fashion and written in standard English?

Reviewer #1: Yes

Reviewer #2: Yes

6. Review Comments to the Author

Reviewer #1: R2:

1. thank you for your patience and response.

Methodoloy:

a. Methodology still need precise effort to be in the frame.

b. Rearange and sumarise the information to avoid rebetition and maintain the integrity of the section.

c. See all the comments in the incloused document. The final description of data which obtained from questionneir must be described in points of one paragraph include all:

1. Personal information

2. Biochemical measurements and their cut-off points

3. Anthrobometric measurements their cut-off points.

4. Sociodemographic and clinical data

5. Life style.

d. Avoid copying the questions and answers of the questionnair. Give only the abstract of your experence.

Reviewer #2: My comments have been adequately addressed.

Congratulations on your efforts!

7. PLOS authors have the option to publish the peer review history of their article (what does this mean?). If published, this will include your full peer review and any attached files.

**Do you want your identity to be public for this peer review?** For information about this choice, including consent withdrawal, please see our Privacy Policy.

Reviewer #1: **Yes: **Nahla Ahmed Mohammed Abderhman

Reviewer #2: No

---

## [Decision Letter · Decision Letter 3]

7 Jun 2024

The burden and care cascade in young and middle-aged patients with Diabetes hypertension comorbidity with abdominal obesity in India: a nationally representative cross-sectional survey

PGPH-D-23-01679R3

Dear Dr. Barzangi,

We are pleased to inform you that your manuscript 'The burden and care cascade in young and middle-aged patients with Diabetes hypertension comorbidity with abdominal obesity in India: a nationally representative cross-sectional survey' has been provisionally accepted for publication in PLOS Global Public Health.

Best regards,

Rajat Das Gupta, M.D.

Academic Editor

Reviewer Comments (if any, and for reference):

Reviewer's Responses to Questions

**Comments to the Author**

1. If the authors have adequately addressed your comments raised in a previous round of review and you feel that this manuscript is now acceptable for publication, you may indicate that here to bypass the “Comments to the Author” section, enter your conflict of interest statement in the “Confidential to Editor” section, and submit your "Accept" recommendation.

Reviewer #1: All comments have been addressed

Reviewer #2: All comments have been addressed

2. Does this manuscript meet PLOS Global Public Health’s publication criteria? Is the manuscript technically sound, and do the data support the conclusions? The manuscript must describe methodologically and ethically rigorous research with conclusions that are appropriately drawn based on the data presented.

Reviewer #1: Yes

Reviewer #2: Yes

3. Has the statistical analysis been performed appropriately and rigorously?

Reviewer #1: Yes

Reviewer #2: Yes

4. Have the authors made all data underlying the findings in their manuscript fully available (please refer to the Data Availability Statement at the start of the manuscript PDF file)?

Reviewer #1: Yes

Reviewer #2: Yes

5. Is the manuscript presented in an intelligible fashion and written in standard English?

Reviewer #1: Yes

Reviewer #2: Yes

6. Review Comments to the Author

Reviewer #1: please re- write this paragraph in methodology:

1. Previously diagnosed patients with DM: Those who were diagnose with high glucose levels or taking medications to lower blood glucose levels.

2. Newly diagnosed patients with DM: Individuals who were not fasting, had RBS levels 200 mg/dl or 126 mg/dl if fasting for 8 hours during the survey, not monitored and noncompliance with hypoglycemic drug to lower glucose level.

3. Newly diagnosed patients with Hypertension: Individuals who had an average of the last two blood pressure readings (BP) 140/90 mmHg on screening during the survey. they were not monitored and with no prescribed medicine to lower BP.

4. Diabetic who use hypotensive drugs.

5. Hypertensive participants who were use medications for treatment of hypertension.

6. return back to comments in the enclosed document.

Reviewer #2: This is quality work!

7. PLOS authors have the option to publish the peer review history of their article (what does this mean?). If published, this will include your full peer review and any attached files.

**Do you want your identity to be public for this peer review?** For information about this choice, including consent withdrawal, please see our Privacy Policy.

Reviewer #1: **Yes: **nahla ahmed Mohammed Abderhman

Reviewer #2: **Yes: **Felix Ayub Barasa
